# Novel *STAG1* Frameshift Mutation in a Patient Affected by a Syndromic Form of Neurodevelopmental Disorder

**DOI:** 10.3390/genes12081116

**Published:** 2021-07-23

**Authors:** Ester Di Muro, Pietro Palumbo, Mario Benvenuto, Maria Accadia, Marilena Carmela Di Giacomo, Sergio Manieri, Rosaria Abate, Maria Tagliente, Stefano Castellana, Tommaso Mazza, Massimo Carella, Orazio Palumbo

**Affiliations:** 1Division of Medical Genetics, Fondazione IRCCS-Casa Sollievo della Sofferenza, 71013 San Giovanni Rotondo (Foggia), Italy; e.dimuro@operapadrepio.it (E.D.M.); p.palumbo@operapadrepio.it (P.P.); m.benvenuto@operapadrepio.it (M.B.); m.carella@operapadrepio.it (M.C.); 2Medical Genetics Service, Hospital “Cardinale G. Panico”, 73039 Tricase (Lecce), Italy; m.accadia@piafondazionepanico.it; 3U.O.C di Anatomia Patologica, AOR Ospedale “San Carlo”, 85100 Potenza, Italy; marilena.digiacomo@ospedalesancarlo.it; 4U.O.C di Pediatria, AOR Ospedale “San Carlo”, 85100 Potenza, Italy; sergio.manieri@ospedalesancarlo.it (S.M.); rosariaabate@virgilio.it (R.A.); dottoressataglientemaria@gmail.com (M.T.); 5Unit of Bioinformatics, Fondazione IRCCS Casa Sollievo della Sofferenza, 71013 San Giovanni Rotondo (Foggia), Italy; s.castellana@css-mendel.it (S.C.); t.mazza@css-mendel.it (T.M.)

**Keywords:** *STAG1*, whole exome sequencing, neurodevelopmental disorders

## Abstract

The cohesin complex is a large evolutionary conserved functional unit which plays an essential role in DNA repair and replication, chromosome segregation and gene expression. It consists of four core proteins, SMC1A, SMC3, RAD21, and STAG1/2, and by proteins regulating the interaction between the complex and the chromosomes. Mutations in the genes coding for these proteins have been demonstrated to cause multisystem developmental disorders known as “cohesinopathies”. The most frequent and well recognized among these distinctive clinical conditions are the Cornelia de Lange syndrome (CdLS, OMIM 122470) and Roberts syndrome (OMIM 268300). STAG1 belongs to the STAG subunit of the core cohesin complex, along with five other subunits. Pathogenic variants in *STAG1* gene have recently been reported to cause an emerging syndromic form of neurodevelopmental disorder that is to date poorly characterized. Here, we describe a 5 year old female patient with neurodevelopmental delay, mild intellectual disability, dysmorphic features and congenital anomalies, in which next generation sequencing analysis allowed us to identify a novel pathogenic variation c.2769_2770del p.(Ile924Serfs*8) in *STAG1* gene, which result to be de novo. The variant has never been reported before in medical literature and is absent in public databases. Thus, it is useful to expand the molecular spectrum of clinically relevant alterations of *STAG1* and their phenotypic consequences.

## 1. Introduction

The cohesin complex mediates sister chromatid cohesion and ensures accurate chromosome segregation, recombination mediated DNA repair, and genomic stability during DNA replication and cell division. Accumulating evidence suggests that cohesin is also involved in regulating chromosomal looping/architecture and gene transcriptional regulation [1]. It consists of four core proteins, SMC1A (OMIM 300040), SMC3 (OMIM 606062), RAD21 (OMIM 606462), and either one of two accessory subunits (STAG1 (OMIM 604358) and STAG2 (OMIM 300826)), and by proteins regulating the interaction between the complex and the chromosome [2]. 

Direct interaction between SMC1A, SMC3, and RAD21 forms a tripartite ring structure that is used to entrap the replicated chromatin during sister chromatid cohesion. STAG1/2 are the core structural components of functional cohesin and critical for the loading of cohesin onto chromatin during mitosis [3,4]. 

Pathologies arising from mutations in the cohesion complex or its regulators are collectively called “cohesinopathies”; multisystem developmental disorders with overlapping clinical features such as distinctive facial dysmorphisms, growth delay, developmental delay/intellectual disability (DD/ID), and limb abnormalities [5]. Among these syndromes, the two best characterized are Cornelia de Lange syndrome (CdLS) (OMIM 122470, 300590, 610759, 614701, 300882) and Roberts syndrome (OMIM 268300). CdLS is an autosomal dominant multiple neurodevelopmental disorder, with an estimated occurrence of 1 in every 10,000. Typical features of CdLS include growth delay with prenatal onset (second trimester), intellectual disability, craniofacial abnormalities, upper limb anomalies, and hirsutism. Additional features include gastroesophageal reflux (GERD), genitourinary malformations, and heart defects [6]. A total of seven genes have been identified in association with CdLS, including *NIPBL* (~50–70%), *SMC1A*, *SMC3*, *BRD4*, *HDAC8* (~4%), *RAD21*, and *ANKRD11*. Roberts syndrome is an autosomal recessive disorder related phenotypically to CdLS, with affected patients demonstrating facial dysmorphisms, limb reduction and growth retardation. Mutations in the gene *ESCO2* were found to be responsible for the Roberts syndrome [7].

To the best of our knowledge, the most important study on individuals carrying alterations of *STAG1* was performed by Lehalle et al. [8], reporting a series of 17 affected patients with *STAG1* alterations, providing the first detailed clinical comparison among cases with this rare condition. Later, three additional de novo *STAG1* pathogenetic variant have been documented [1]. Taken together, these variants included seven microdeletions, and thirteen single nucleotide variants (ten missense, three frameshift).

Here, we describe a 5 year old female patient carrier of a novel frameshift variant in the *STAG1* gene. The variant, determined to be de novo, is useful in expanding the mutational spectrum of *STAG1*. In addition, this case shares common features previously reported as well as expands on the phenotypes associated with *STAG1* variations.

## 2. Materials and Methods

### 2.1. Genomic DNA Extraction and Quantification

This family provided written informed consent to molecular testing and to the full content of this publication. This study was conducted in accordance with the 1984 Declaration of Helsinki and its subsequent revisions. The conservation and the use of biological samples for scientific purposes were approved by the Casa Sollievo della Sofferenza Hospital ethics committee (protocol no. 177CE). Peripheral blood samples were taken from both the proband and her parents, and genomic DNA was isolated by using Bio Robot EZ1 (Quiagen, Solna, Sweden). The quality of DNA was tested on 1% electrophorese agarose gel, and the concentration was quantified by Nanodrop 2000 C spectrophotometer (Thermo Fisher Scientific, Waltham, MA, USA).

### 2.2. Whole-Exome Sequencing

Proband DNA was analyzed by Whole Exome sequencing (WES) by using SureSelect Human Clinical Research Exome V6 (Agilent Technologies, Santa Clara, CA, USA) following manufacturer instructions. This is a combined shearing-free transposase-based library prep and target-enrichment solution, which enables comprehensive coverage of the entire exome. This system enables a specific mapping of reads to target deep coverage of protein-coding regions from RefSeq, GENCODE, CCDS, and UCSC Known Genes, with excellent overall exonic coverage and increased coverage of HGMD, OMIM, ClinVar, and ACMG targets. Sequencing was performed on a NextSeq 500 System (Illumina, San Diego, CA, USA) by using the High Output flow cells (300 cycles), with a minimum expected coverage depth of 70x. All variants obtained from WES were called by means of the HaplotypeCaller tool of GATK ver. 3.58 [9] and were annotated based on frequency, impact on the encoded protein, conservation, and expression using distinct tools, as appropriate (ANNOVAR, dbSNP, 1000 Genomes, EVS, ExAC, ESP, KAVIAR, and ClinVar) [10,11,12,13,14], and retrieving pre-computed pathogenicity predictions with dbNSFP v 3.0 (PolyPhen-2, SIFT, MutationAssessor, FATHMM, LRT and CADD) [15] and evolutionary conservation measures.

Next, variant prioritization was performed. Firstly, variants described as benign and likely benign were excluded. Then, remaining variants were classified based on their clinical relevance as pathogenic, likely pathogenic, or variant of uncertain significance according to following criteria: (i) nonsense/frameshift variant in genes previously described as disease-causing by haploinsufficiency or loss-of-function; (ii) missense variant located in a critical or functional domain; (iii) variant affecting canonical splicing sites (i.e., ±1 or ±2 positions); (iv) variant absent in allele frequency population databases; (v) variant reported in allele frequency population databases, but with a minor allele frequency (MAF) significantly lower than expected for the disease (<0.002 for autosomal recessive disease and <0.00001 for autosomal dominant disease); (vi) variant predicted and/or annotated as pathogenic/deleterious in ClinVar and/or LOVD. 

The resulting putative pathogenic variants were confirmed by Sanger sequencing in both the proband and the parents’ DNA. PCR products were sequenced by using BigDye Terminator v1.1 Sequencing Kit (Applied Biosystems, Foster City, CA, USA) and ABI Prism 3100 Genetic Analyzer (Thermo Fisher Scientific). The clinical significance of the identified putative variants was interpreted according to the American College of Medical Genetics and Genomics (ACMG) [16] guidelines. Variant analysis was carried out considering the ethnicity of the patient.

Nucleotide variants nomenclature follows the format indicated in the Human Genome Variation Society (HGVS, http://www.hgvs.org) recommendations (accessed on 1 March 2021). 

## 3. Results

### 3.1. Clinical Description

The proband was a 5-year-old Caucasian girl born to healthy non-consanguineous parents with a syndromic form of intellectual disability of unknown etiology. There was no family history of congenital anomalies or intellectual disability (ID)/neurodevelopmental disorders (NDD). She was born after a normal full-term pregnancy, with a birth weight of 3350 g, length of 52 cm. Occipitofrontal circumference (OFC) was not reported. There were no remarkable events during the perinatal period. Her developmental milestones were delayed; she raised her head at 1 year, stood up and walked at 26 months, spoke her first words at 18 months. In addition, the parents reported an episode of uveitis at 4 years old, angiomas of the midline (one of the nasal columella removed at 2 years), frequent respiratory tract infections, late dental eruption (at 16 months), recurrent juvenile idiopathic arthritis of the oligoarticular type that began at 18 months, and hypocalcemia. On physical examination at 5 years old, her height was 114 cm (75–90th percentile), weight was 20 kg (50–75th percentile), and OFC was 49 cm (10–25th percentile). Her dysmorphic facial features included relative microcephaly, a prominent forehead, deep set eyes, asymmetrical nasal tip, raised ear lobes, mild micrognathia, widely spaced teeth (Figure 1). She also had pectus excavatum, clinodactyly of the 5th finger and persistence of finger fetal pads. Electroencephalography (EEG) anomalies of likely intercritical significance in the middle right front were observed while electrocardiogram and echocardiogram were unremarkable as well as vision and hearing evaluations resulted both normal. Conventional G-banded karyotype analysis showed a normal female karyotype (46,XX); chromosomal microarray analysis (CMA) performed by using CytoScan HD Array from Affymetrix (Thermos Fisher Scientific, Waltham, MA, USA), revealed no abnormality. 

### 3.2. Genetic Analysis

The WES allowed us to identify a novel variation c.2769_2770del p.(Ile924Serfs*8) in the exon 26 of *STAG1* gene. The variant was detected with a depth coverage greater than 150x, and with good quality scores (Phread quality ˃ 3000 and genotype quality = 99). This frameshift variant causes a premature stop codon and was absent in 1000G, ExAC, dbSNP, EP6500, and our in-house controls. Bioinformatics details are reported in Table 1. 

Co-segregation analysis was investigated in the family. The primers were designed using Primer3.0 (http://bioinfo.ut.ee/primer3-0.4.0) (accessed on 1 January 2021) (*STAG1*, exon 26, Forward primer: TGACACCAACCTATTCTTTATCTCA; *STAG1*, exon 26, Reverse Primer: CACTGACAAAATGATATAAAATGACC) and the polymerase chain reaction (PCR) was performed under standard conditions. The PCR product (298 bp) was sequenced on an ABI 3500xL DNA Analyzer (Applied Biosystems, Foster City, CA, USA). Parental DNA analysis showed that it is a de novo event (Figure 2). 

According to the ACMG guidelines, the variant detected was classified as likely pathogenic and reported in the Leiden Open Variation Databases (LOVD) (https://databases.lovd.nl/shared/variants/0000789637) (accessed on 1 June 2021).

## 4. Discussion

*STAG1* encodes for a subunit of evolutionarily conserved cohesin complex, which plays a crucial role in the control of chromosome segregation during cell division. In fact, it is required for the cohesion of sister chromatids, and therefore, ensures the proper distribution of genetic material to daughter cells [7]. Besides this canonical role, recent data demonstrated the involvement of cohesin complex in gene transcription, regulation, and DNA repair and replication. Therefore, mutations in genes encoding these proteins have the potential to damage all these processes, with important functional repercussions for several cellular mechanisms. Pathologies arising from mutations in the cohesin complex or its regulators are collectively called “cohesinopathies” [5]. 

*STAG1* belongs to the STAG subunit of the core cohesin complex, along with four other subunits. It was shown that STAG1 contributes to the chromatin architecture, and studies of the transcriptomes of STAG1-null and wild-type mouse embryonic fibroblasts revealed transcriptional changes in the STAG1-null cells. STAG proteins also appear to be necessary for normal development, with STAG1 knockout mice showing developmental defects and embryonic lethality [17]. These data suggest a specific role of STAG1 in gene regulation, essential for embryonic development. 

In this study we report the case of a girl with a neurodevelopmental disorder (DD/ID), craniofacial dysmorphisms, pectus excavatum, hands with clinodactyly of the 5th finger and persistence of fetal pads. Aiming to reveal the underlying genetic cause of this syndromic clinical manifestation, we employed WES, which was able to identify a novel heterozygous frameshift *STAG1* variant. This variant was not found in the parents, which was consistent with de novo inheritance.

To the best of our knowledge, so far 20 patients have been reported in the literature and all *STAG1* variants described for these cases have been associated with neurodevelopmental delay (intellectual disability and developmental delay), which may be accompanied by a plethora of additional clinical features and remarkable variable expressivity [1,8]. In detail, the most frequently observed features are intellectual disability/developmental delay (20/20, 100%), feeding difficulties (9/20, 45%), seizures (9/20, 45%), autism spectrum disorders (8/20, 40%) and growth (prenatal and/or postnatal) retardation (6/20, 30%). In addition, although it is currently not possible to delineate a specific recurrent pattern for the dysmorphic facial features observed useful in performing/suspecting a clinical diagnosis based on this evidence, some somatic traits are more frequent such us deep set eyes (14/20, 70%), a wide mouth (13/20, 65%), high nasal bridge (7/20, 35%), thin eyebrows (8/20, 40%), and widely spaced teeth (4/20, 20%). Finally, less frequently documented signs are cryptorchidism (2/10, 20%), scoliosis (2/20, 10%), and congenital heart defects (2/20, 10%). The present case shares some of these phenotypes, including growth retardation, ID/DD, and facial dysmorphisms (relative microcephaly, deep set eye, wide mouth, widely spaced teeth). Our patient also showed clinical features rarely documented in patients with *STAG1*-related phenotypes including micrognathia (which had been reported in only one patient) and, more importantly, showed pectus excavatum, clinodactyly of the 5th finger and persistence of finger fetal pads which have never been documented before. These findings could be additional clinical features typical of the STAG1-syndrome. Obviously, more patients and genotype–phenotype correlation studies are needed to corroborate these data and to better delineate the clinical spectrum of this rare cohesinopathy. 

From a molecular point of view, the variant c.2769_2770del p.(Ile924Serfs*8) detected in our patient was never reported before in medical literature, is absent in public databases, and is thus useful to expand the molecular spectrum of pathogenic alterations of *STAG1*. 

Furthermore, a careful observation of the clinical and molecular features documented in the affected individuals, including ours, suggests that the etiopathogenetic mechanism may be the same, namely *STAG1* haploinsufficiency. In fact, although to date very heterogeneous patients from a genetic point of view (carriers of *STAG1* deletions, frameshift variants, missense) have been identified, and there does not seem to be a mutational hot spot for the *STAG1* gene which has instead emerged for other clinical conditions such as Schinzel-Giedion syndrome (SGS, OMIM 269150) [18], all the patients described share specific clinical features. Therefore, in agreement with other authors [8], we suggest that the clinical manifestations and severity of the STAG1-syndrome does not depend on the nature/type of the gene variant but arises through transcriptional dysregulation due to depletion/defects of cohesin complex. Further functional studies on cellular and/or animal models, and analysis of gene expression profiles are needed to reinforce this emerging evidence on the etiopathogenetic mechanisms and to delineate the key events responsible for the onset of this rare cohesinopaty.

## 5. Conclusions

In this work, we report on a Caucasian girl with a novel heterozygous *STAG1* gene variant and clinical features associated. Our case displayed a phenotype consistent with previous studies of *STAG1* variants, and enriched the current clinical phenotypic data of this emergent and still poorly characterized syndrome. Further exome sequencing of patients with ID of unknown cause is needed to continue to identify *STAG1* variants and to allow us to better delineate the associated clinical features and molecular mechanisms driving the onset of the disease. 

## Figures and Tables

**Figure 1 genes-12-01116-f001:**
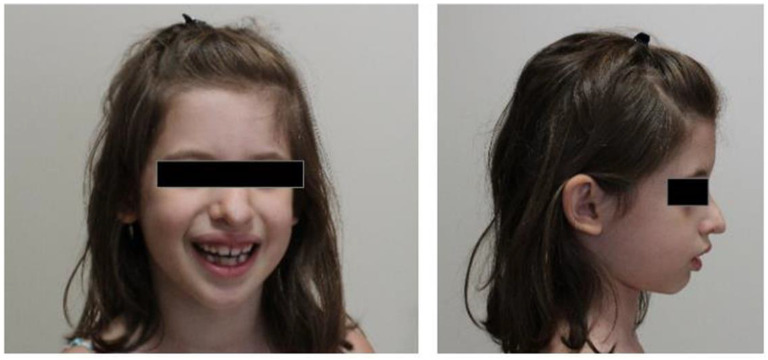
Facial phenotype of the patient.

**Figure 2 genes-12-01116-f002:**
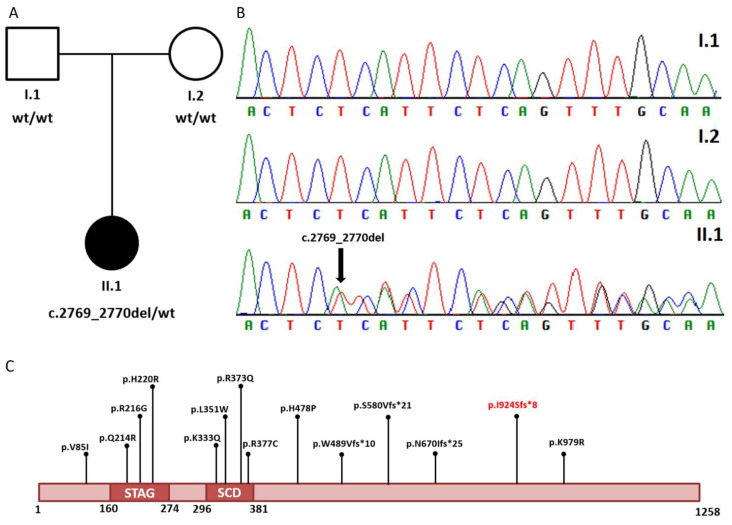
(**A**) Pedigree of the family displaying the de novo onset of the variant. Filled and unfilled circles/squares represent affected and unaffected individuals, respectively. (**B**) Electropherograms of the proband (II.1) and her parents (I.1, I.2). The variant identified in the proband is indicated by black arrow (**C**) Schematic representation of STAG1 protein and reported variants [1]. The variant identified here is indicated in red (SCD: stromalin conserved domain).

**Table 1 genes-12-01116-t001:** Characteristics of the variant identified in the *STAG1* gene.

Chr	Start	End	Reference Allele	Alternative Allele	Genotype	Gene	Exonic Function	Nucleotide Change	Amino Acid Change	**GnomAD Exome** **Allele Count**	**DbSNP ID**	**ExAC ALL** **Allele Count**
3	136082225	136082227	TGA	T	Het	*STAG1*	Frameshift substitution	c.2768_2769del	p.(Ile924Serfs*8)	N.R.	N.R.	N.R.

Het = heterozygous; N.R. = not reported.

## Data Availability

The data presented in this study are openly available in ArrayExpress database (https://www.ebi.ac.uk/arrayexpress/) under the accession number E-MTAB-10811.

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
