# Peer review of "Novel STAG1 Frameshift Mutation in a Patient Affected by a Syndromic Form of Neurodevelopmental Disorder"

_genes, 2021, doi:10.3390/genes12081116_

Round 1
Reviewer 1 Report
The novel pathogenic variation c.2769_2770del p.(Ile924Serfs*8) in the STAG1 gene is well described. The associated neurodevelopmental delay, mild intellectual disability, dysmorphic features and congenital anomalies give direction to studies of the clinical features correlated with STAG1 variants.
Minor corrections are required :
Lines 56 - 57 please pay attention to the font size
Line 74 Add more information to " This family " as for example is given at the beginning of the Results, in lines 120 - 121. " The proband ............till...........unknown etiology. "
Line 77 "his" should be "her"
Line 138 please pay attention to the font size
Last line of page 6 and first line of page 7 please pay attention to the font size
On page 8 the last two lines of the conclusions (they basically are repeats). Please the authors should make up their minds, which conclusion they prefer. In any case, it is advised to split the sentence in two. Ending the first part at "STAG1 variants." with a period. And then make a transition with the start of the second sentence, where the authors can present the implications.
Author Response
Author's Reply to the Review Report (Reviewer 1)
The novel pathogenic variation c.2769_2770del p.(Ile924Serfs*8) in the STAG1 gene is well described. The associated neurodevelopmental delay, mild intellectual disability, dysmorphic features and congenital anomalies give direction to studies of the clinical features correlated with STAG1 variants.
Minor corrections are required:
- Lines 56 - 57 please pay attention to the font size
ANSWER: thank you for suggestion, we adapted the font
- Line 74 Add more information to " This family " as for example is given at the beginning of the Results, in lines 120 - 121. " The proband ............till...........unknown etiology. "
ANSWER: thank you very much for the suggestion but we think that it is inappropriate to describe in detail the family in the section “2.1. Genomic DNA Extraction and Quantification” while is more appropriate, as previously done for others paper submitted and accepted on your esteemed journal, to report in details all the informations regarding the family history in the section “3.1. Clinical Description”.
- Line 77 "his" should be "her"
ANSWER: done
- Line 138 please pay attention to the font size
ANSWER: we adapted the font
- Last line of page 6 and first line of page 7 please pay attention to the font size
ANSWER: we adapted the font
On page 8 the last two lines of the conclusions (they basically are repeats). Please the authors should make up their minds, which conclusion they prefer. In any case, it is advised to split the sentence in two. Ending the first part at "STAG1 variants." with a period. And then make a transition with the start of the second sentence, where the authors can present the implications.
ANSWER: sorry for the inconvenient, we forgot to remove the last one. As suggested, we retained only one of the sentences and in particular we prefer: “Further exome sequencing of patients with ID of unknown cause is needed to continue to identify STAG1 variants and to allow us to better delineate the clinical features associated and the molecular mechanisms driving the onset of the disease.”

Reviewer 2 Report
The authors identified a novel and de novo cohesin gene variant that is associated with a neurodevelopmental presentation. The study provides more evidence for STAG1 to be involved in a syndromic disease.
The introduction provides a good picture of the context.
The methods are sound, standard and they are described with enough detail and follow a logical flow.
In the discussion, an appropiate and useful description on the clinical presentation of cases associated with STAG1 variants is provided.
Minor concerns:
There is a missing reference for Roberts syndrome (page 2, line 61)
Some minor text editing needs to be done.
Major concerns:
In the discussion:
"transcriptomes of STAG1-null and wild-type mouse embryonic fibroblasts revealed transcriptional changes in the STAG1-null cells." this paragraph doesn't add any information and even more, leaves more questions open about what are those genes that are dysregulated. Please add any relevant information about the changes, given that even just the deletion of a gene could produce gene expression changes, STAG1 or any other, therefore more specific information will be useful.
Author Response
Author's Reply to the Review Report (Reviewer 2)
The authors identified a novel and de novo cohesin gene variant that is associated with a neurodevelopmental presentation. The study provides more evidence for STAG1 to be involved in a syndromic disease.
The introduction provides a good picture of the context.
The methods are sound, standard and they are described with enough detail and follow a logical flow.
In the discussion, an appropiate and useful description on the clinical presentation of cases associated with STAG1 variants is provided.
Minor concerns:
There is a missing reference for Roberts syndrome (page 2, line 61)
ANSWER: we added the missing reference as number 7.
Some minor text editing needs to be done.
ANSWER: as suggested, we performed text editing, in particular regarding the font size of some periods. In addition, we corrected the name of the first author.
Major concerns:
In the discussion: "transcriptomes of STAG1-null and wild-type mouse embryonic fibroblasts revealed transcriptional changes in the STAG1-null cells." this paragraph doesn't add any information and even more, leaves more questions open about what are those genes that are dysregulated. Please add any relevant information about the changes, given that even just the deletion of a gene could produce gene expression changes, STAG1 or any other, therefore more specific information will be useful.
ANSWER: we understand the point of view of the reviewer but our aim was not to add any informations to what we already known. Basically, with this paragraph we wanted to cite a paper in which has been clearly demonstrated an discussed the important role of STAG1 gene in gene expression regulation and the impact that its deregulation have on the normal functioning of the most important biological processes involved in embryonic development. All these aspects emerges and are discussed in detail in the paper cited as reference [17] (Arruda, NL.; Carico, ZM .; Justice, M .; Liu, YL .; Zhou, J .; Stefan, HC .; Dowen, JM Distinct and overlapping roles of STAG1 and STAG2 in cohesin localization and gene expression in embryonic stem cells. Epigenetics Chro-matin. 2020 Aug 10; 13 (1): 32.).
